# Toward Sustainable Development: The Causes and Consequences of Organizational Innovation

**Li-Min Chuang [1] and Yu-Po Lee [2],***

1   Department of International Business, Chang Jung Christian University, Tainan 711301, Taiwan;
    liming@mail.cjcu.edu.tw
2   The Ph.D. Program in Business and Operations Management, College of Management,
    Chang Jung Christian University, Tainan 711301, Taiwan
*   Correspondence: 109d00107@mail.cjcu.edu.tw

**Abstract:** As society continues to evolve, environmental contextual factors continue to change. The primary purpose of this study is to investigate the relationship between organizational innovation and individual, organizational, and environmental context variables, as well as the impact of organizational innovation on the performance of an organization. This study will investigate the incorporation of relevant aspects of environment, society, and governance into organizational innovation, and investigate its influencing factors on innovation. The information electronics industry based at Hsinchu Science Park was selected to gather data for this study. Overall, the number of valid questionnaires recovered was 138, with an effective recovery rate of 45.25% (138/305). The findings of this study are in support of environmental and organizational variables having the largest explanatory power for organizational innovation, while individual creativity was found to have less of a correlation with organizational innovation. This study has expanded and continued to make breakthroughs and contributions in studies on ESG and sustainability-oriented organizational innovation.

**Keywords:** organizational innovation; sustainable development; sustainability-oriented organizational innovation

## 1. Introduction

### 1.1. The Background and Needs of the Study

The development of a sustainable economy may involve encouraging systems that minimize consumption, designing systems that maximize social and environmental benefits, designing closed-loop systems that aim not to throw any waste into the environment, designing systems that emphasize functionality and performance but not product ownership, and creating systems that collaborate and share. The above-mentioned possibilities require a fundamental change in the way organizations operate, and organizational innovation helps to re-conceptualize the company's objectives and value creation, thus providing the required changes and reflecting on values. A study by Aouadi and Marsat [1] also suggested that the management of companies that integrate ESG strategies are better able to secure competitive advantage, improve operational efficiency, maintain reputation, reduce waste, and ultimately enhance shared value and sustainability with stakeholders. A study by Harsanto et al. [2] has reviewed the literature on sustainability-oriented innovation in social enterprises, which found that process and organizational innovation, such as business model transformation and stakeholder management, can help social enterprises to increase their social impact. Mercedes Rubio-Andrés and Abril [3] discussed that sustainability-oriented innovation is closely related to organizational values.

Research into organizational innovation often considers the question of what causes organizations to become more innovative, with researchers exploring the antecedents to the advancement or inhibition of innovation in an organization. Several factors have already been identified, such as Wolfe [4] stating that individual, organizational, and environmental

factors are variables which affect organizational innovation. Scott [5] developed a set of determinants of innovative behavior in organizations, including leadership, individual attributes, work teams, and an innovative work mentality. Higgins [6] proposed the concept of the innovation equation, suggesting that creativity and organizational culture are important factors affecting innovation. King and Anderson [7] suggest that the factors influencing organizational innovation can be subdivided into people, structure, climate and culture, and the environment. Many studies on the influencing factors of organizational innovation demonstrate that individual, organizational, and environmental context variables are all capable of explaining organizational innovation [8–13]. This study will further investigate the relationship between organizational innovation and individual, organizational, and environmental context variables, as well as the impact of organizational innovation on the performance of an organization.

Based on the above, it can be seen that business operations inevitably face the impact of environment, society, and corporate governance. In order to maintain a sustainable business, companies must engage in sustainability-oriented innovation. However, what the cause and consequence factors of organizational innovation are, and which factors are more important are the core motivations for this study.

### 1.2. Purpose of the Study

As society continues to evolve and environmental contextual factors continue to change, the paradigm of corporate management has changed from 'the pursuit of maximum profit' to 'improving the world through ESG'. ESG stands for environmental protection (E, environment), social responsibility (S, social), and corporate governance (G, governance), and is a specific policy for corporations to implement sustainable business practices. This study will investigate the incorporation of relevant aspects of ESG into organizational innovation and investigate its influencing factors on innovation. Niu et al. [14] have previously investigated the relationship between digital leadership, ESG management, organizational innovation, and sustainability. However, as the environment changes, corporations are increasingly adopting ESG methods and regulations, and while the previous literature on the integration of ESG into business models has shown it is worth increasing and investing in, this literature on ESG and business model innovation is still only at the analytical framework stage. Most studies still only investigate whether ESG investments are beneficial to financial performance. Aldowaish et al. [15] identified two principal axes in ESG studies: socially responsible investment (SRI), which focuses on investment from an ESG perspective, and sustainable development (SD), which investigates ESG from the perspective of a company's operations. To date, most of the literature has focused on socially responsible investment, with only a few studies incorporating ESG into corporate operations. Although corporations are increasingly adopting ESG methods and regulations, little is known about how to incorporate ESG into business models [15]. The core concept of integrating ESG into business models refers to integration into the following four aspects of a business model: value proposition, value creation, value delivery, and value acquisition. The past literature on the integration of sustainability with business models has taken many interesting investigative approaches, including defining the characteristics of a business model [16,17], analyzing the model framework [18], developing business model schema [19], discussing how to virtualize the business model [20], and modelling sustainable business models [21]. This study will focus on the development of sustainability-oriented innovation in order to analyze the causes and consequences of organizational innovation of the above-mentioned research approaches. We will further investigate the integration of ESG into corporate operations, as well as individual, organizational, and environmental context variables, in order to investigate the relationship between organizational innovation and the performance of an organization.

### 1.3. Contribution of the Study

Innovation-related research is often limited to technical or technological aspects. Research on innovation often varies greatly in content and direction. Although research results are accumulating, the interpretation and clarification of the meaning of organizational innovation is still inadequate, especially in the light of the significant environmental, social, and corporate governance pressures that companies are currently facing for sustainable development. It is particularly important to integrate these concepts and to explore the causes and consequences of organizational innovation. Although there are many empirical studies exploring the influencing factors of organizational innovation and the relation between organizational innovation and organizational performance, no consistent conclusions have been obtained, or the development model is incomplete. This study therefore builds on this foundation and explores further the research gap in sustainability-oriented organizational innovation. Through the empirical analysis of this study, the perspectives and connotations of sustainability-oriented organizational innovation will be extended and enriched, making this study valuable both in the academic field and in practical application.

## 2. Theoretical Background and Hypothesis Development

For scholars conducting organizational research, the question of how to improve organizational performance through the introduction of innovation has always been an important issue. Although some scholars have researched the characteristics of innovative organizations, the findings have been so inconsistent that it has been difficult to construct a theory of organizational innovation. Organizational innovation is a complex construct, and so to define it only in terms of products, processes, or any other sole indicator would inevitably be biased, and not provide a complete picture. As such, many scholars tend to support a multi-perspective approach to organizational innovation, as there is a belief that previous scholars placed too much emphasis on technological innovation, overlooking aspects of administrative innovation. Technological innovation refers to the technology involved in products, services, and the manufacturing process. It is directly related to the fundamental operations of an organization and encompasses both product and process innovations [8,9,22,23]. Administrative innovation instead involves the organizational structures and management processes of an organization, and so, administrative innovation is only indirectly related to the fundamental operations of an organization but is directly related to the management of an organization [8,9,22]. Both technological and administrative innovation put focus onto the fundamental operations of an organization and can be distinguished through the balance of and interaction between the technological and managerial systems of an organization. This study also supports the multi-perspective of scholars in stating that organizational innovation comprises both technological innovation (including products, processes, and facilities) and administrative innovation (including systems, policies, programs, and services). Organizational innovation is defined as the adoption of a concept or behavior that is new to the organization, which could include a new product, service, technology, or management practice. This is a relatively consistent definition from previous studies [9,22,24–28].

### 2.1. The Causes of Organizational Innovation

The research of Saleh and Brem [29] reviewed the relation between creativity and sustainability and cites studies of how creativity affects sustainability in different fields, among which individual creativity often affects sustainable consumption behavior [30–32]. As for the factors influencing organizational innovation, much of the literature suggests that individual, organizational, and environmental contextual variables have explanatory power [8–11]. Amabile [10] emphasized the importance of individual motivation, task-related skills, and creative thinking skills for innovation, based on the three components of creativity theory, in which five stages of organizational innovation were proposed and the influence of individual creative variables on organizational innovation was regarded. Woodman et al. [33] expanded their theoretical model of organizational creativity with

a model of individual behavioral interaction. In this model, individual creativity is a function of antecedents, cognitive styles and abilities, personality, motivational factors, and knowledge, and it also interacts with social contextual factors to contribute to organizational innovation. Kanter [34] and Amabile [10] both pointed out that the variable of individual creativity is an important variable affecting organizational innovation, and in addition, personality traits are an important factor in the development of individual creativity. Accordingly, Hypotheses 1 and 4 are introduced in this study.

**H1:** The higher the degree of creativity presents within the individuals of an organization, the higher the level of overall organizational innovation.

Maqdliyan and Setiawan [35] studied antecedents and consequences of public sector organizational innovation and compiled the antecedents affecting organizational innovation, namely internal control system [36], organizational culture [37], and transformational leadership [38]. Many papers on the influences of organizational innovation show that individual, organizational, and contextual variables all have explanatory power; however, the organizational variables have the highest explanatory power. The more relevant organizational variables according to the past literature include organizational characteristics, organizational culture, organizational climate, and organizational structural design [8–11]. Damanpour [9] used meta-analysis to examine the relation between "organizational structure" and "organizational innovation" and found "centralization" has a significant negative correlation with organizational innovation, "formalization" is not associated with organizational innovation, and "specialization" is significantly positively correlated with organizational innovation. Accordingly, Hypotheses 2 and 4 are developed in this study.

**H2:** The degree of organizational innovation will vary with different organizational structures.

Much of the literature on the influencing factors of organizational innovation shows that individual, organizational, and contextual variables all have explanatory power, and environmental factors are also an important interference factor for organizational innovation. Environmental factors can interfere with the impact of research and development on team characteristics and innovation performance (product and process innovation). If there is a high degree of uncertainty and frequent changes in the environment and too much emphasis is placed on communication and cooperation between teams, decisions cannot be made quickly and correctly, resulting in lower product and process innovation performance. Damanpour [39] adopted the definition of multi-organizational innovation and found in his research that when the environmental uncertainty is high, there is a positive effect on the relation between organizational complexity and organizational innovation and the relation between organizational size and organizational innovation, indicating that when the environmental uncertainty is high, there will be a high positive correlation in the relation between organizational complexity and organizational innovation and the relation between organizational size and organizational innovation. Accordingly, Hypotheses 3 and 4 are developed in this study.

**H3:** The degree of organizational innovation will vary with different environmental characteristics.

**H4:** The variables which impact organizational innovation (individual, organizational, and environmental variables) all have significant effects on organizational innovation, with organizational and environmental variables having the greater significant effects.

### 2.2. The Consequences of Organizational Innovation

Yamin et al. [40] examined the relation between innovation indicators and performance, and it was found that organizational innovation (management innovation, technological innovation, and product innovation) is significantly related to performance. The development of organizational innovation is an urgent issue for modern enterprises. The study by Niu et al. [14] examined the impact of digital leadership and ESG management on organizational innovation and sustainability. The study also mentioned that organi-

zational innovation has a positive impact on performance, competitive advantage, and sustainability [41]. Accordingly, Hypotheses 5 are developed in this study.

**H5:** The higher the levels of organizational innovation, the better the performance of an organization.

### 3. Research Design

*3.1. Research Framework*

It can be seen that business operations inevitably face the impact of environment, society, and corporate governance. Companies must engage in sustainability-oriented innovation. However, what the cause and consequence factors of organizational innovation are, and which factors are more important are the core motivations for this study. Based on the aforementioned research background, research objectives, and related literature, the following research framework, shown in Figure 1, was established, and the relationship between the definitions of operational variables and the variables themselves are explained below.

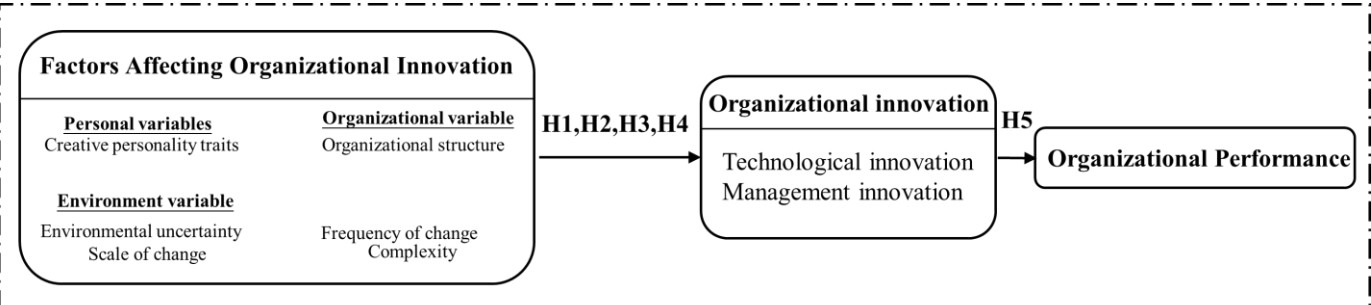

**Figure 1.** Research framework.

3.1.1. Industries

The industries studied are from the Taiwanese information and electronics sector, including the semiconductor industry, the computer and peripherals industry, the communications industry, and the optoelectronics industry.

3.1.2. Factors Influencing Organizational Innovation

Many studies on the factors influencing organizational innovation show that individual, organizational, and environmental context variables all have explanatory power, and this study further examines which of these influencing factors is more important. We divide the factors affecting organizational innovation into three categories:

(1)　Individual variables: investigating the creative personality traits present in individual staff members.
(2)　Organizational variables: investigating the characteristics of the organization (staff numbers, year of establishment, the organization's life cycle, amount of capital, types of capital) and the structure of the organization (centralization, formalization, organizational complexity).
(3)　Environmental variables: mainly investigating the environment facing the organization, which is divided into environmental uncertainty, scale of change, frequency of change, and environmental complexity.

3.1.3. Organizational Innovation

In the empirical research section, organizational innovation is measured using our Organizational Innovation Scale, and said scale is comprised of seven dimensions: product innovation, process innovation, creative work environments, marketing innovation, organizational characteristic innovation, organizational system innovation, and strategy innovation.

### 3.1.4. Organizational Performance

The index for measuring organizational performance in this study includes both financial and non-financial indicators, including return on equity (ROE), earnings per share (EPS), sales growth rate, and market share.

### *3.2. Measuring Tools*

According to the research framework diagram in Figure 1, the empirical research section of this study will investigate the relationship between organizational innovation variables (individual, organizational, and environmental), organizational innovation, and organizational performance. The measurement tools used in this study are explained below.

### 3.2.1. Scale of Creative Personality Traits in Individuals

This measurement is based off of the Creative Personality Scale (CPS) developed by Gough in 1979. The scale includes 30 attributes, of which 18 were positively associated with creativity (question numbers: 2, 4, 6, 10, 12, 13, 14, 15, 16, 18, 19, 21, 22, 23, 24, 25, 27, 30), and the remaining 12 were negatively associated with creativity. Respondents were scored with 1 point for each positive association, and −1 point for each negative association. After completing all 30 questions the scores were totaled, and the higher the score, the higher the level of creative personality traits. However, the results of the study showed that there were many respondents with overall negative scores. In order to make analysis more convenient, those with a total innovation score of 0 or less were classified as "low creative personality traits" ($n = 56$), those with scores of 1–5 were classified as "moderate creative personality traits" ($n = 59$), and those with scores of 6+ were classified as "high creative personality traits" ($n = 23$).

### 3.2.2. Organizational Characteristics

Organizational characteristics are measured by factors such as number of employees, year of establishment, organizational life cycle, amount of capital, and types of capital.

### 3.2.3. Organizational Structure

According to the previous literature, the more relevant organizational variables include organizational characteristics, organizational culture, organizational climate, and the design of an organization's structure [8–11]. In this study, the measurement and classification of organizational structure was performed using Damanpour's 7-point Likert scale [9], which measures employee perceptions of their organization. The responses are divided into seven categories: strongly disagree, somewhat disagree, neutral, somewhat agree, agree, and strongly agree. The scores are 1, 2, 3, 4, 5, 6, and 7 for each category, respectively. Within the 'Centralization' section, there are eight questions total, and only the seventh is positively associated with innovation. Within the 'Formalization' section, there are seven questions total, all positively associated with innovation, and in the 'Complexity' section, there is a single question, also positively associated with innovation.

### 3.2.4. Environmental Characteristics

The content of this scale is based on the definitions from Gomez-Mejia et al. [42], who used a 7-point Likert scale to measure the perceptions of information and electronics industry executives on the environment of their organization. The responses were divided into the seven categories of strongly disagree, somewhat disagree, neutral, somewhat agree, agree, and strongly agree. The scores are 1, 2, 3, 4, 5, 6, and 7 for each category, respectively. Gomez-Mejia et al. [42] identified four environmental factors which affect organizational innovation: degree of uncertainty, frequency of change, scale of change, and environmental complexity.

### 3.2.5. Organizational Innovation Scale

The Organizational Innovation Scale is based on the definitions from Chuang and Tsai [43], and also uses the 7-point Likert scale to measure employee perceptions of their organization. The scale is divided into seven responses: strongly disagree, somewhat disagree, neutral, somewhat agree, agree, and strongly agree. The scores are 1, 2, 3, 4, 5, 6, and 7 for each category, respectively. The scale consists of seven components: product innovation, process innovation, creative work environments, marketing innovation, organizational characteristic innovation, organizational system innovation, and strategy innovation.

### 3.2.6. Organizational Performance

The index for measuring organizational performance in this study includes both financial and non-financial indicators, including return on equity (ROE), earnings per share (EPS), sales growth rate, and market share. Among which, the first two are objective indicators, and the latter two are indicators based on subjective perceptions.

### 3.3. Data Collection

The information electronics industry based at Hsinchu Science Park was selected to gather data for this study. The classification of the industries according to the Science Park is as follows: the total number of companies across the integrated circuit, optoelectronics, communications, and computer and peripheral industries is 305 (132 in the integrated circuit industry, 59 in the optoelectronics industry, 61 in the communications industry, and 53 in the computer and peripherals industry). In order to ensure appropriate respondents completed the questionnaire, only 1 was distributed to each company, making 305 the total number of distributed questionnaires. A total of 48 valid questionnaires were recovered in the first stage of the study (15.74% effective recovery rate). In the second stage, a total of 300 questionnaires were redistributed to companies who did not respond in stage one, and researchers made telephone reminders, resulting in the collection of 90 valid questionnaires (30% effective recovery rate). Overall, the number of valid questionnaires recovered was 138, with an effective recovery rate of 45.25% (138/305). A general sampling method was used for purposive sampling. The judging criterion is to select a middle-level executive from each company who is suitable for answering this questionnaire. In order to better understand the views of experts in the high-tech industry, both practical and academic, on the importance of organizational innovation constructs and indicators, and to validate the validity and reliability of the scale and model, this study will target the domestic information and electronics industry, and we will select companies and send questionnaires to R&D, human resources, and marketing executives for empirical study. A sample of this study is shown in Table 1.

**Table 1.** Sample structure.

| Basic Features | Classification | Sample Size (*n* = 138) | Percentage (%) | Cumulative Percentage (%) |
|---|---|---|---|---|
| Number of employees | Less than 200 people | 40 | 29.0 | 29.0 |
| | 201–500 people | 36 | 26.1 | 55.1 |
| | 501–1000 people | 15 | 10.9 | 65.9 |
| | 1001–2000 people | 21 | 15.2 | 81.2 |
| | 2001–3000 people | 17 | 12.3 | 93.5 |
| | More than 3001 people | 9 | 6.5 | 100.0 |
| Years of establishment | Under 5 years | 44 | 31.9 | 31.9 |
| | 5–10 years | 30 | 21.7 | 53.6 |
| | Over 10 years | 64 | 46.4 | 100.0 |

**Table 1.** *Cont.*

| Basic Features | Classification | Sample Size (*n* = 138) | Percentage (%) | Cumulative Percentage (%) |
|---|---|---|---|---|
| Organizational life cycle | bud | 8 | 5.8 | 5.8 |
| | growing up | 72 | 52.2 | 58.0 |
| | Mature | 47 | 34.1 | 92.0 |
| | decline | 7 | 5.1 | 97.1 |
| | regeneration | 4 | 2.9 | 100.0 |
| Capital amount | Below 100 million | 29 | 21.0 | 21.0 |
| | 100–600 million | 55 | 39.9 | 60.9 |
| | More than 600 million | 54 | 39.1 | 100.0 |
| Capital attribute | Domestic capital | 108 | 78.3 | 78.3 |
| | Sino-foreign joint venture | 27 | 19.6 | 97.8 |
| | foreign capital | 3 | 2.2 | 100.0 |
| Industry | Semiconductor industry | 34 | 24.6 | 24.6 |
| | Optoelectronics | 28 | 20.3 | 44.9 |
| | Communications | 33 | 23.9 | 68.8 |
| | Computer and peripheral industry | 43 | 31.2 | 100.0 |

## 4. Empirical Results

This section will analyze the correlation between individual, organizational, and environmental variables and organizational innovation. The following is an analysis of the effect of each variable on organizational innovation, and the empirical results in cases when all three variables were found to have an effect on organizational innovation at the same time. The analysis can be summarized as follows.

### 4.1. Analysis of the Correlation between Individual Variables and Organizational Innovation

Pearson correlation analysis was used to illustrate the relationship between individual variables (creative personality traits) and organizational innovation. As shown in Table 2, the results of the study show that creative personality traits are significantly and positively correlated with all aspects of organizational innovation. Product innovation and marketing innovation have a relatively high correlation with creative personality traits, while the correlation coefficient between technological innovation and creative personality traits was higher than that with management innovation. Hypothesis 1 is therefore supported. Therefore, the higher the levels of creativity present within the individuals of an organization, the higher the level of overall organizational innovation.

**Table 2.** Analysis of correlation coefficient between creative personality traits and organizational innovation and each dimension (*n* = 138).

| | Product Innovation | Process Innovation | Creativity Working Environment | Marketing Innovation | Organizational Characteristic Innovation | Organization System Innovation | Strategic Innovation | Organizational Innovation | Management Innovation | Technological Innovation |
|---|---|---|---|---|---|---|---|---|---|---|
| Creative personality traits | 0.598 *** | 0.510 *** | 0.357 *** | 0.598 *** | 0.575 *** | 0.571 *** | 0.573 *** | 0.734 *** | 0.644 *** | 0.649 *** |

Note: *** $p < 0.001$.

### 4.2. Analysis of the Correlation between Organizational Variables and Organizational Innovation

Pearson correlation analysis and multiple regression analysis were used to illustrate the relationship between organizational variables (centralization, formalization, organizational complexity) and organizational innovation, as shown in Tables 3 and 4. In summary, Hypothesis 2 is only partially supported. Therefore, the higher the degree of centralization and formalization, the higher the degree of organizational innovation. Complexity has no significant effect on the degree of organizational innovation. In order to further understand the impact of organizational structure on organizational innovation, a complex regression analysis was conducted on the impact of the three variables (centralization, formalization,

and organizational complexity) on organizational innovation, as shown in Table 4. The results show that for organizational innovation, centralization and formalization have a positive significant effect, while complexity has a negative significant effect. This means that the higher the centralization or formalization, or the lower the complexity of an organization, the higher the degree of organizational innovation.

**Table 3.** Organizational structure, environmental characteristics, and organizational innovation and correlation coefficient analysis among each dimension (*n* = 138).

| | Product Innovation | Process Innovation | Creative Work Environment | Marketing Innovation | Organizational Characteristic Innovation | Organization System Innovation | Strategic Innovation | Organizational Innovation | Management Innovation | Technological Innovation |
|---|---|---|---|---|---|---|---|---|---|---|
| Uncertainty | −0.462 *** | −0.580 *** | −0.657 *** | −0.563 *** | −0.707 *** | −0.749 *** | −0.647 *** | −0.745 *** | −0.752 *** | −0.565 *** |
| Frequency of change | 0.504 *** | 0.566 *** | 0.569 *** | 0.586 *** | 0.655 *** | 0.738 *** | 0.565 *** | 0.703 *** | 0.695 *** | 0.592 *** |
| Change scale | 0.256 ** | 0.246 ** | 0.268 ** | 0.370 *** | 0.352 *** | 0.438 *** | 0.339 *** | 0.378 *** | 0.382 *** | 0.285 ** |
| Complexity | 0.120 | 0.333 *** | 0.355 *** | 0.639 *** | 0.424 *** | 0.413 *** | 0.359 *** | 0.403 *** | 0.430 *** | 0.213 * |
| Centralization | 0.127 | 0.086 | 0.297 *** | 0.175 * | 0.218 * | 0.177 * | 0.155 | 0.240 ** | 0.256 ** | 0.128 |
| Formalized | 0.280 ** | 0.509 *** | 0.419 *** | 0.455 *** | 0.485 *** | 0.532 *** | 0.495 *** | 0.524 *** | 0.529 *** | 0.400 *** |
| Complication | −0.048 | −0.035 | −0.111 | 0.023 | −0.100 | 0.024 | −0.51 | −0.064 | −0.064 | −0.049 |

Note: * $p < 0.05$; ** $p < 0.01$; *** $p < 0.001$.

**Table 4.** Regression analysis of organizational structure on organizational innovation.

| Dependent Variable: Organizational Innovation | | |
|---|---|---|
| **Independent Variable** | **Coefficient** | ***t*-Value** |
| Constant | 104.678 | 4.007 *** |
| Centralization | 1.317 | 2.782 ** |
| Formalized | 5.079 | 8.510 *** |
| Complication | −8.397 | −3.089 ** |
| F | | 28.362 *** |
| $R^2$ | | 0.388 |
| Adj $R^2$ | | 0.375 |
| DW | | 1.96 |

Note: ** $p < 0.01$; *** $p < 0.001$.

*4.3. Analysis of the Correlation between Environmental Variables and Organizational Innovation*

Pearson correlation analysis and multiple regression analysis were used to illustrate the relationship between environmental variables (uncertainty, frequency of change, scale of change, environmental complexity) and organizational innovation, as shown in Tables 3 and 5. Table 3 shows that uncertainty is significantly and negatively correlated with all aspects of organizational innovation and innovation itself, and the correlations between uncertainty and organizational system innovation, characteristic innovation, and creative work environments are all relatively high. The correlation between administrative innovation and formalization is higher than that with technological innovation, but the relationship is negatively correlated. It can be seen from the above evidence that the greater the uncertainty in an organization's environment, the lower the degree of innovation will be. The atmosphere around innovation will also be affected, with management systems especially becoming more conservative. Summarizing the above findings, Hypothesis 3 is, for the most part, supported. In summary, the higher the levels of uncertainty, the lower the levels of organizational innovation. The higher the frequency of change, and the greater the scale of change and degree of complexity, the higher the levels of organizational innovation.

**Table 5.** Regression analysis of environmental characteristics on organizational innovation.

| Dependent Variable: Organizational Innovation | | |
|---|---|---|
| **Independent Variable** | **Coefficient** | ***t*-Value** |
| Constant | 277.194 | 9.945 *** |
| Uncertainty | −6.233 | −6.813 *** |
| Frequency of change | 7.464 | 5.032 *** |
| Change scale | 0.656 | 3.437 ** |
| Complexity | 0.328 | 2.358 ** |
| F | | 58.133 *** |
| $R^2$ | | 0.636 |
| Adj $R^2$ | | 0.625 |
| DW | | 2.22 |

Note: ** $p < 0.01$; *** $p < 0.001$.

To better understand the effects of environmental characteristics on organizational innovation, this study takes the four variables of uncertainty, frequency of change, scale of change, and environmental complexity and conducts a complex regression analysis on their effects on organizational innovation. This can be seen in Table 5. The results of the research show that frequency of change, scale of change, and complexity all have a positive significant impact on organizational innovation, while uncertainty has a negative significant impact on innovation. The higher a corporation's frequency of change, scale of change, or complexity, or the lower the level of uncertainty, then the degree of organizational innovation within the corporation will increase.

*4.4. Analysis of the Effects of Individual, Organizational, and Environmental Variables on Organizational Innovation*

In order to investigate the effects of individual, organizational, and environmental variables on organizational innovation, as well as the interaction between the variables, and the explanatory power of the variables on organizational innovation, this study conducts stepwise multiple regression analysis based on the explanatory power of the independent variables on the dependent variables to progressively measure the predicted effects of each variable, and the unique cause and effect relationships between independent variables. This study uses stepwise analysis, first using forward stepwise regression to incorporate the best predictor variables. After each predictor variable is integrated, backward stepwise regression is used to test the predictor variables in the equation. If any predictor variables are found to not be significant, they are removed, and the process continues until all predictor variables in the equation are significant, and all variables that have been removed are not significant. The results of stepwise regression are presented in Table 6. The results show that through stepwise regression, five statistically significant variables were selected (Model 5): uncertainty, frequency of change, centralization, formalization, and complexity, all of which are classified as organizational or environmental variables, with an explained variance of 69.2%. Among these, uncertainty and complexity both have a significant negative correlation with organizational innovation. Interestingly, the first important predictor variables that were selected for the model, uncertainty (Model 1) and frequency of change (Model 2), are both environmental variables, which suggests that of the factors affecting organizational innovation, environmental factors may have the greater explanatory power. The next important predictor variables that were selected for the model are centralization (Model 3), formalization (Model 4), and organizational complexity (Model 5), with these all being classified as organizational variables. This means that organizational variables are also important explanatory variables affecting organizational innovation but have less explanatory power than environmental variables. Compared to environmental and organizational variables, the impact of the individual

variable of creative personality traits does not have a statistically significant effect on organizational innovation.

**Table 6.** Stepwise regression analysis of the impact of individual variables, organizational variables, and environmental variables on organizational innovation.

| Model | Variable | Select Variable | Unnormalized Coefficient | | Standardized Coefficient | *t*-Value | *p*-Value | Adj R$^2$ | Adj R$^2$ the Amount of Change |
| --- | --- | --- | --- | --- | --- | --- | --- | --- | --- |
| | | | Estimated Value of B | Standard Error | Beta Distribution | | | | |
| 1 | (constant) Uncertainty | Uncertainty | 399.839 −9.544 | 9.452 0.733 | −0.745 | 42.300 −13.013 | 0.000 *** 0.000 *** | 0.551 | 0.551 |
| 2 | (constant) Uncertainty frequency of change | frequency of change | 288.328 −6.384 7.737 | 22.208 0.884 1.421 | −0.498 0.376 | 12.983 −7.223 5.445 | 0.000 *** 0.000 *** 0.000 *** | 0.629 | 0.78 |
| 3 | (constant) Uncertainty frequency of change centralization | centralization | 242.524 −5.863 8.214 1.045 | 26.283 0.875 1.388 0.343 | −0.457 0.399 0.157 | 9.227 −6.702 5.917 3.046 | 0.000 *** 0.000 *** 0.000 *** 0.003 ** | 0.651 | 0.022 |
| 4 | (constant) Uncertainty frequency of change centralization Formalized | Formalized | 186.105 −4.964 7.574 1.178 1.407 | 31.924 0.904 1.367 0.337 0.477 | −0.387 0.368 0.177 0.172 | 5.830 −5.493 5.539 3.500 2.950 | 0.000 *** 0.000 *** 0.000 *** 0.001 ** 0.004 ** | 0.670 | 0.019 |
| 5 | (constant) Uncertainty frequency of change centralization Formalized complication | complication | 198.724 −4.842 7.255 0.947 1.888 −4.649 | 31.843 0.890 1.351 0.345 0.511 1.971 | −0.378 0.352 0.142 0.231 −0.131 | 6.241 −5.440 5.370 2.741 3.692 −2.359 | 0.000 *** 0.000 *** 0.000 *** 0.007 ** 0.000 *** 0.020 * | 0.681 | 0.011 |

Model 1 Predictor Variables: (constant), Uncertainty; Model 2 Predictor Variables: (constant), Uncertainty, frequency of change; Model 3 Predictor Variables: (constant), Uncertainty, frequency of change, centralization; Model 4 Predictor Variables: (constant), Uncertainty, frequency of change, centralization, Formalized; Model 5 Predictor Variables: (constant), Uncertainty, frequency of change, centralization, Formalized, complication; Dependent variable: Organizational innovation; * $p < 0.05$; ** $p < 0.01$; *** $p < 0.001$.

Based on the above findings, and in order to further test the influencing factors on organizational innovation and figure out which has the greatest explanatory power, hierarchical multiple regression will be carried out for the following analysis. Hierarchical multiple regression analysis is used when a researcher has a clear theoretical basis for the dividing and sorting of variables in advance, rather than, as in the case of the above-mentioned stepwise regression, using statistical magnitude as the foundation for selecting predictor variables. This study conducted hierarchical multiple regression to test the results. The three groups of variables (individual, organizational, and environmental) were put into the regression formula, with the order of input based on the importance of the variables taken from the stepwise multiple regression (environmental first, then organizational, then finally individual variables), with the results shown in Table 7. The results show that the first group of variables (environmental variables, Model 1) had an explained variance of 63.6%, the second group (environmental and organizational variables, Model 2) had a significant rise in explained variance ($\Delta R^2 = 6.1\%$), and the third group (environmental, organizational, and individual variables, Model 3) showed no significant rise in explained variance ($\Delta R^2 = 1.4\%$). From these results, it can be seen that environmental variables have the greatest explanatory power for organizational innovation. This is followed by organizational variables, and then individual variables, which have no significant explanatory power for organizational innovation.

In terms of the standardized β values in Table 7, Model 1 shows that environmental uncertainty and frequency of change demonstrate a significant negative relationship and positive relationship, respectively, with organizational innovation (β = −0.486, $p < 0.001$; β = 0.362, $p < 0.001$). Scale of change has a positive but not statistically significant correlation with organizational innovation (β = 0.027), and environmental complexity similarly has a negative but not statistically significant correlation (β = −0.023).

**Table 7.** Results of hierarchical regression: relationship between individual variables, organizational variables, environmental variables, and organizational innovation.

| Independent Variable | Model 1 (Normalized β Value) | Model 2 (Normalized β Value) | Model 3 (Normalized β Value) |
|---|---|---|---|
| Environment variable | | | |
| environmental uncertainty | −0.486 *** | −0.365 *** | −0.364 *** |
| frequency of change | 0.362 *** | 0.348 *** | 0.347 *** |
| scale of change | 0.027 | 0.077 | 0.077 |
| Complexity | −0.023 | −0.045 | −0.045 |
| Organizational variable | | | |
| centralization | | 0.160 ** | 0.161 ** |
| Formalized | | 0.234 ** | 0.234 ** |
| complication | | −0.131 * | −0.130 * |
| Individual variables | | | |
| creative personality traits | | | 0.007 |
| Adj $R^2$ | 62.5% *** | 68.0% *** | 69.7% *** |
| Δadj $R^2$ | | 5.5% *** | 1.9% |

Note: * $p < 0.05$; ** $p < 0.01$; *** $p < 0.001$.

Model 2 shows that environmental uncertainty and frequency of change demonstrate a significant negative relationship and positive relationship, respectively, with organizational innovation (β = −0.365, $p < 0.001$; β = 0.348, $p < 0.001$). Scale of change has a positive but not statistically significant correlation with organizational innovation (β = 0.077), and environmental complexity similarly has a negative but not statistically significant correlation (β = −0.045). Centralization and formalization both have a positive significant relationship with organizational innovation (β = 0.160, $p < 0.01$; β = 0.234, $p < 0.01$), while organizational complexity has a negative significant correlation (β = −0.131, $p < 0.05$).

Model 3 shows that environmental uncertainty and frequency of change demonstrate a significant negative relationship and positive relationship, respectively, with organizational innovation (β = −0.364, $p < 0.001$; β = 0.347, $p < 0.001$). Scale of change has a positive but not statistically significant correlation with organizational innovation (β = 0.077), and environmental complexity similarly has a negative but not statistically significant correlation (β = −0.045). Centralization and formalization both have a positive significant relationship with organizational innovation (β = 0.161, $p < 0.01$; β = 0.234, $p < 0.01$), while organizational complexity has a negative significant correlation (β = −0.130, $p < 0.05$). Creative personality traits are positively but not significantly correlated with organizational innovation (β = 0.007).

Based on the results and analysis from Tables 6 and 7, the most important factors affecting organizational innovation were found to be environmental and organizational variables, therefore supporting Hypothesis 4. Many previous studies have found that individual, organizational, and environmental variables all have explanatory power [2,4,18,19], who has previously conducted a series of research into organizational innovation, places great emphasis on individual creativity on organizational innovation. Additionally, many studies into organizational innovation have found that individual, organizational, and environmental variables all have explanatory power, but have found that organizational variables exhibit the greatest explanatory power. There are also many scholars who believe that environmental factors are the most important disruptive variable to organizational innovation. The findings of this study are in support of environmental and organizational variables having the largest explanatory power, while individual creativity was found to have less of a correlation with organizational innovation, which is completely different to the findings of Amabile [10]. From the above results, we have found that individuals can change quickly to adapt to rapid environmental changes while organizations are slower to change. The stakes are such that the organization directly affects the environment and

the individuals within. If a company's profits have mostly been greater than their costs, then it will be difficult for ideas of change and innovation to take hold. However, if there are major changes to the environment surrounding the company causing progress to go downhill, this is when thoughts of making drastic changes may arise. The environment, speed of change, and external pressures are all related. If the situation facing the environment is urgent, stressful, oppressive, and demanding, the speed of change among people within the environment will also increase. If the environment is strong, then the power given to the organization will be relatively large, which in turn stimulates innovation. Thus, the organization acts as a large organism with the people within as a variable and the environment acting as a background force, and so if there is relatively little pressure and change in the environment, the organization will also not change. Organizational innovation is therefore often a response to rapid environmental changes, and the greater the speed and scope of change, the more innovative corporations must be in response. In uncertain circumstances, corporations will instead keep a relatively conservative attitude to innovation. Organizations must adapt to rapid environmental changes. Appropriate centralization and formalization will foster a degree of organizational innovation, while highly complex organizations are likely to be less innovative. The results of this study show that individual creativity does not have a significant impact on organizational innovation, and the reason for this may be the difference in the research approach compared to Amabile [10], and that considering it alongside other variables may lead to a decrease in its explanatory power. There is still a significant positive relationship between creativity and organizational innovation, and so managers should not neglect this variable. They should examine the environment, adjust the organization, and increase creativity, so that the overall capability for organizational innovation can be enhanced through the mutual cooperation of individuals, organizations, and the environment.

In a highly competitive business environment, the degree of organizational innovation is critical to the performance of the organization. Yamin et al. [40] examined the relationship between innovation indicators and performance and found that organizational innovation (management innovation, technological innovation, and product innovation) is significantly related to performance. The development of organizational innovation is an urgent issue for modern enterprises. Accordingly, in this study, the organizational innovation level and ranking of the study sample by using the developed organizational innovation model is calculated, as shown in Table 8. Since the model for measuring organizational innovation developed in this study includes subjective indicators and objective indicators, and it is obtained through the process of standardized Z transformation multiplied by weights, the results are more rigorous. As for organizational performance, the indicators for measuring organizational performance in this study include financial and non-financial indicators, namely the four indicators of return on assets (ROE), earnings per share (EPS), company sales growth rate, and company market share. The calculation of organizational performance is also obtained by summing the values of the four indicators through the process of standardized Z-transformation, then ranking them according to their magnitude, as shown in Table 8. By using the formula of Spearsman's correlation coefficient analysis [Spearsman's correlation coefficient = $1 - 6\frac{\sum di^2}{n(n^2-1)}$; $n = 138$], the correlation coefficient of this study is 0.396 ($p$-value = 0.007). Therefore, the analysis of the correlation between organizational innovation and organizational performance in this study concludes that organizational innovation and organizational performance present a significant positive correlation. That is, the higher the degree of organizational innovation, the higher the organizational performance may be. Hypothesis 5 is supported. Based on the above analysis, most of the scholars and this study agree that organizational innovation is positively correlated with organizational performance. However, the causal relationship between them has not been verified in this study, and even some scholars believe that the ability to innovate is only possible with high performance, and the causal relationship exists, but it is not concluded in this study.

**Table 8.** Analysis of Spearsman's correlation coefficient between organizational innovation and organizational performance.

| No. | Organizational Innovation Ranking | Organizational Performance Ranking | di | di$^2$ |
|---|---|---|---|---|
| 1 | 117 | 44 | −73 | 5329 |
| 2 | 54 | 40 | −14 | 196 |
| 3 | 119 | 138 | 19 | 361 |
| 4 | 137 | 45 | −92 | 8464 |
| 5 | 115 | 109 | −6 | 36 |
| 6 | 23 | 110 | 87 | 7569 |
| 7 | 67 | 17 | −50 | 2500 |
| 8 | 9 | 92 | 83 | 6889 |
| 9 | 73 | 8 | −65 | 4225 |
| 10 | 19 | 15 | −4 | 16 |
| 11 | 134 | 111 | −23 | 529 |
| 12 | 25 | 127 | 102 | 10,404 |
| 13 | 62 | 46 | −16 | 256 |
| 14 | 92 | 93 | 1 | 1 |
| 15 | 16 | 18 | 2 | 4 |
| 16 | 37 | 35 | −2 | 4 |
| 17 | 28 | 128 | 100 | 10,000 |
| 18 | 109 | 129 | 20 | 400 |
| 19 | 84 | 130 | 46 | 2116 |
| 20 | 64 | 131 | 67 | 4489 |
| 21 | 101 | 47 | −54 | 2916 |
| 22 | 125 | 112 | −13 | 169 |
| 23 | 5 | 33 | 28 | 784 |
| 24 | 93 | 48 | −45 | 2025 |
| 25 | 15 | 49 | 34 | 1156 |
| 26 | 128 | 36 | −92 | 8464 |
| 27 | 11 | 50 | 39 | 1521 |
| 28 | 10 | 51 | 41 | 1681 |
| 29 | 17 | 52 | 35 | 1225 |
| 30 | 34 | 43 | 9 | 81 |
| 31 | 30 | 1 | −29 | 841 |
| 32 | 61 | 19 | −42 | 1764 |
| 33 | 89 | 39 | −50 | 2500 |
| 34 | 90 | 94 | 4 | 16 |
| 35 | 49 | 20 | −29 | 841 |
| 36 | 138 | 120 | −18 | 324 |
| 37 | 1 | 2 | 1 | 1 |
| 38 | 106 | 79 | −27 | 729 |

**Table 8.** *Cont.*

| No. | Organizational Innovation Ranking | Organizational Performance Ranking | di | di² |
|---|---|---|---|---|
| 39 | 111 | 21 | −90 | 8100 |
| 40 | 33 | 95 | 62 | 3844 |
| 41 | 24 | 96 | 72 | 5184 |
| 42 | 130 | 97 | −33 | 1089 |
| 43 | 129 | 80 | −49 | 2401 |
| 44 | 112 | 98 | −14 | 196 |
| 45 | 50 | 6 | −44 | 1936 |
| 46 | 71 | 81 | 10 | 100 |
| 47 | 96 | 16 | −80 | 6400 |
| 48 | 70 | 53 | −17 | 289 |
| 49 | 12 | 7 | −5 | 25 |
| 50 | 29 | 54 | 25 | 625 |
| 51 | 18 | 55 | 37 | 1369 |
| 52 | 51 | 22 | −29 | 841 |
| 53 | 68 | 23 | −45 | 2025 |
| 54 | 113 | 82 | −31 | 961 |
| 55 | 87 | 87 | 0 | 0 |
| 56 | 105 | 37 | −68 | 4624 |
| 57 | 118 | 121 | 3 | 9 |
| 58 | 40 | 132 | 92 | 8464 |
| 59 | 80 | 99 | 19 | 361 |
| 60 | 132 | 122 | −10 | 100 |
| 61 | 91 | 56 | −35 | 1225 |
| 62 | 79 | 57 | −22 | 484 |
| 63 | 63 | 58 | −5 | 25 |
| 64 | 14 | 100 | 86 | 7396 |
| 65 | 52 | 59 | 7 | 49 |
| 66 | 42 | 41 | −1 | 1 |
| 67 | 83 | 101 | 18 | 324 |
| 68 | 47 | 60 | 13 | 169 |
| 69 | 100 | 113 | 13 | 169 |
| 70 | 103 | 102 | −1 | 1 |
| 71 | 110 | 123 | 13 | 169 |
| 72 | 107 | 103 | −4 | 16 |
| 73 | 55 | 83 | 28 | 784 |
| 74 | 4 | 24 | 20 | 400 |
| 75 | 95 | 25 | −70 | 4900 |
| 76 | 85 | 61 | −24 | 576 |
| 77 | 7 | 62 | 55 | 3025 |
| 78 | 75 | 26 | −49 | 2401 |

**Table 8.** *Cont.*

| No. | Organizational Innovation Ranking | Organizational Performance Ranking | di | di$^2$ |
|---|---|---|---|---|
| 79 | 74 | 9 | −65 | 4225 |
| 80 | 31 | 63 | 32 | 1024 |
| 81 | 94 | 64 | −30 | 900 |
| 82 | 32 | 10 | −22 | 484 |
| 83 | 69 | 114 | 45 | 2025 |
| 84 | 20 | 27 | 7 | 49 |
| 85 | 122 | 124 | 2 | 4 |
| 86 | 88 | 115 | 27 | 729 |
| 87 | 127 | 116 | −11 | 121 |
| 88 | 99 | 65 | −34 | 1156 |
| 89 | 2 | 11 | 9 | 81 |
| 90 | 46 | 104 | 58 | 3364 |
| 91 | 48 | 66 | 18 | 324 |
| 92 | 3 | 3 | 0 | 0 |
| 93 | 114 | 133 | 19 | 361 |
| 94 | 77 | 67 | −10 | 100 |
| 95 | 27 | 68 | 41 | 1681 |
| 96 | 108 | 28 | −80 | 6400 |
| 97 | 98 | 69 | −29 | 841 |
| 98 | 36 | 70 | 34 | 1156 |
| 99 | 6 | 117 | 111 | 12,321 |
| 100 | 59 | 118 | 59 | 3481 |
| 101 | 22 | 71 | 49 | 2401 |
| 102 | 43 | 134 | 91 | 8281 |
| 103 | 56 | 42 | −14 | 196 |
| 104 | 72 | 84 | 12 | 144 |
| 105 | 76 | 85 | 9 | 81 |
| 106 | 123 | 125 | 2 | 4 |
| 107 | 53 | 72 | 19 | 361 |
| 108 | 66 | 34 | −32 | 1024 |
| 109 | 35 | 29 | −6 | 36 |
| 110 | 120 | 73 | −47 | 2209 |
| 111 | 116 | 105 | −11 | 121 |
| 112 | 45 | 30 | −15 | 225 |
| 113 | 126 | 12 | −114 | 12,996 |
| 114 | 38 | 13 | −25 | 625 |
| 115 | 21 | 14 | −7 | 49 |
| 116 | 65 | 106 | 41 | 1681 |
| 117 | 131 | 89 | −42 | 1764 |
| 118 | 39 | 74 | 35 | 1225 |

**Table 8.** *Cont.*

| No. | Organizational Innovation Ranking | Organizational Performance Ranking | di | di² |
|---|---|---|---|---|
| 119 | 121 | 107 | −14 | 196 |
| 120 | 102 | 90 | −12 | 144 |
| 121 | 81 | 38 | −43 | 1849 |
| 122 | 57 | 75 | 18 | 324 |
| 123 | 60 | 31 | −29 | 841 |
| 124 | 86 | 4 | −82 | 6724 |
| 125 | 13 | 76 | 63 | 3969 |
| 126 | 58 | 32 | −26 | 676 |
| 127 | 8 | 5 | −3 | 9 |
| 128 | 44 | 77 | 33 | 1089 |
| 129 | 135 | 108 | −27 | 729 |
| 130 | 104 | 78 | −26 | 676 |
| 131 | 41 | 86 | 45 | 2025 |
| 132 | 133 | 126 | −7 | 49 |
| 133 | 124 | 135 | 11 | 121 |
| 134 | 78 | 136 | 58 | 3364 |
| 135 | 82 | 91 | 9 | 81 |
| 136 | 136 | 119 | −17 | 289 |
| 137 | 26 | 88 | 62 | 3844 |
| 138 | 97 | 137 | 40 | 1600 |

Note: Spearsman's correlation coefficient $= 1 - 6 \frac{\sum di^2}{n(n^2-1)}$; $n = 138$.

### 4.5. Qualitative Analysis and Research Results

4.5.1. Qualitative Analysis

In order to validate the findings and strengthen the credibility of the interpretations, in-depth interviews and the focus group technique were used. A total of six experts, three from academia and three from practice, were recruited to conduct focus group discussions, in which the results of the statistical analysis of the study were discussed and all agreed that the findings of the study were reasonable, and the interpretation of the findings was appropriate.

The majority of the interviewees agreed that overall organizational innovation must be based on the premise that "the individual, the organization and the environment must work together." As the environment changes rapidly, individuals can change quickly to adapt to the environment, while organizations change more slowly. The organization has a direct impact on the environment and the individual. The environment and speed are related to external pressures. Organizational innovation is often a response to rapid change in the environment, and the speed and magnitude of change often forces companies to respond with innovative approaches. However, when uncertainty is high, companies are more conservative in their approach to innovation. The best way to improve the overall innovation capability of an organization is to examine the environment, adjust the organization, and enhance creativity. Companies should carefully assess the environment and make adjustments to their organizations and individuals.

4.5.2. Research Results

The results of this study are summarized in Table 9.

**Table 9.** Comprehensive summary analysis table of this study.

| Hypothesis | Research Method | Research Results |
|---|---|---|
| H1 | 1. Pearson correlation analysis.<br>2. Focus group technique. | Supported |
| H2 | 1. Multiple regression analysis.<br>2. Focus group technique. | Supported |
| H3 | 1. Multiple regression analysis.<br>2. Focus group technique. | Supported |
| H4 | 1. Stepwise regression analysis.<br>2. Hierarchical multiple regression methods.<br>3. Focus group technique. | Supported |
| H5 | 1. Spearsman's correlation analysis.<br>2. Focus group technique. | Supported |

## 5. Conclusions and Recommendation

### 5.1. Conclusions

According to the findings presented in Table 9, the research hypotheses H1 to H5 are supported. This means that the causes of organizational innovation, including individual variables, have a significant impact on organizational innovation, which is consistent with previous studies [20,30–32]. Additionally, organizational variables also have a significant influence on organizational innovation, aligning with previous research [9,36–38]. Moreover, the consequences of organizational innovation, including its impact on performance, are significant, which is consistent with previous studies [40].

Furthermore, this study extends the research focus on sustainable development by integrating the ESG perspective into organizational innovation and exploring the relationships between its antecedents and consequences. This is in line with the research orientations of studies [18,20,21]. Subsequent researchers can transform the findings of this study into three analytical frameworks for business model innovation: business creation, delivery, and value capture. This can contribute to the development of sustainable organizational innovation or business model innovation research models, facilitating empirical studies by future researchers.

Many previous studies into the factors affecting organizational innovation have found that individual, organizational, and environmental variables all have explanatory power. Amabile [10], who has previously conducted a series of research into organizational innovation, places great emphasis on individual creativity on organizational innovation. Additionally, many studies into organizational innovation have found that individual, organizational, and environmental variables all have explanatory power, but have found that organizational variables exhibit the greatest explanatory power. There are also many scholars who believe that environmental factors are the most important disruptive variable to organizational innovation. The findings of this study are in support of environmental and organizational variables having the largest explanatory power for organizational innovation, while individual creativity was found to have less of a correlation with organizational innovation, which is completely different to the findings of Amabile [10].

In interpreting the above results, we believe that the possible reasons for them are as follows: integrating an ESG perspective for analysis, corporations with environmental values (E) are based on environmental awareness and strong corporation governance

capabilities, and through the process of productization, they are consistently engaged in advanced responses to the market. Through product design and continuous testing and amendments, new markets can be developed, and equipment purchased. This creates advantages against competitors that are difficult to match, and aids in the gain of differentiated and monopolistic profits. At the same time, corporations oriented towards environmental value (E) will focus on environmental protection and robust corporate governance capabilities, allowing them to develop new markets and attract new customers. By combining with partners with a similar philosophy (e.g., suppliers, cross-industry alliances) and conducting mutual research, creativity can be developed, and a circular economy which can be reused, generating value innovation and models for long-term stable revenue. Corporations oriented towards social value (S) will build new relations with customers and restructure the service processes for product use due to their focus on public welfare and robust corporate governance. This is so that customers can access integrated services, and a safe and harmonious society can be balanced with a space to make profits. Corporations oriented towards social value (S) focus on public welfare and robust corporate governance. Through new channels, they use new technological capabilities to provide customized products and services, shape new business models, and greatly reduce the costs of business transactions. At the same time, corporations oriented towards social value (S) focus on public welfare and robust corporate governance, and through new channels they simplify processes and raise the customer retention rate. Consumers can choose differentiated products or services in accordance with their needs, and the corporation can greatly reduce transaction costs. Summarizing the above points about environmental value-oriented corporations, they are based on awareness around environmental protection and strong corporate governance capabilities (G), use new technology to develop new supply platforms, offer completely new products and services, and form new partnerships. Consumers can choose differentiated products or services in accordance with their needs, and the corporation can gain profits from price differentiation, while at the same time can reduce costs on a large scale due to the widespread use of the Internet. The above analysis conforms with the research results which showed that environmental and organizational variables have a relatively large effect on and explanatory power for organizational innovation.

The findings of this study deepen the perspective reached by Evans et al. [44], Harsanto et al. [2], and Rubio-Andrés and Abril [3]. Through the method of a questionnaire, ESG was integrated into the study of the antecedents and consequences of organizational innovation and the testing of relevant hypotheses. This study has expanded and continued to make breakthroughs and contributions in studies on ESG and sustainability-oriented organizational innovation.

### 5.2. Limitation and Recommendation

(1) Individual, organizational, and environmental variables all have explanatory power. However, the findings of this study are in support of environmental and organizational variables having the largest explanatory power, while individual creativity was found to have less of a correlation with organizational innovation. On the whole, the corporations who participated in our research believed that individual, organizational, and environmental variables acting in coordination are required for organizational innovation to develop. The environment can change rapidly, and individuals can also quickly adapt to those changes, while organizations are much slower to change, and organizations can directly affect the environment and individuals, respectively. The environment, speed of change, and external pressures are all related. If the situation facing the environment is urgent, stressful, oppressive, and demanding, the speed of change among people within the environment will also increase. If the environment is strong, then the power given to the organization will be relatively large, which in turn stimulates innovation. Thus, the organization acts as a large organism with the people within as a variable and the environment acting as a background

force, and so if there is relatively little pressure and change in the environment, the organization will also not change. Organizational innovation is often a response to rapid environmental changes, and the greater the speed and scope of change, the more innovative corporations must be in response. In uncertain circumstances, corporations will instead keep a relatively conservative attitude towards innovation. Examining the overall environment, making adjustments within the organization, and enhancing creativity is the only way for organizations to improve their overall capacity for organizational innovation. Corporations should carefully assess the work environment, and then adjust the organization and individuals accordingly.

(2) Although we attempted to ensure the company selection process for this study was rigorous, due to the limitations on company cooperation, it was not possible to ensure all of the benchmark innovative companies took part. Additionally, only one respondent was interviewed per company, meaning it was easy for their subjective views to influence them and affect the results of the study. It is recommended that subsequent researchers select a greater number of representative case companies and interview multiple executives per company or increase the level of the executives interviewed in order to make the research more rigorous.

(3) The organizational innovation measurement model developed in this study is inclined towards the measurement of the two major systemic components of organizational innovation: technological and administrative innovation. Subsequent researchers may wish to add the concepts of industrial innovation, social innovation, and national innovation systems to further investigate their relationship with and impact on organizational innovation, which will supplement or expand the content and components of the research model, allowing for a more comprehensive and rigorous measurement model of organizational innovation. Additionally, subsequent researchers could add the proliferation of innovation as an approach to their research. Integration with the three studies by Wolfe [4] will make the conclusions of studies into organizational innovation more comparable and complete.

(4) Although the research model constructed for this study has been empirically analyzed for some high-tech companies in Taiwan, the sample size is still not sufficient. Therefore, there is still room to strengthen the sample size of the organizational innovation measurement model developed in this study. Subsequent researchers can expand the sample of the study for empirical analysis to re-test the reliability and validity of this measurement model and make it more rigorous.

(5) The measurement of individual creativity is not easy, and the use of creative personality scale in this study may result in some biases. It is suggested that subsequent researchers may use other measures to strengthen the generalization ability and avoid possible problems.

**Author Contributions:** Conceptualization, L.-M.C.; methodology, L.-M.C.; validation, L.-M.C. and Y.-P.L.; formal analysis, L.-M.C.; investigation, Y.-P.L.; data curation, L.-M.C. and Y.-P.L.; writing—original draft preparation, Y.-P.L.; writing—review and editing, L.-M.C. All authors have read and agreed to the published version of the manuscript.

**Funding:** This research received no external funding.

**Institutional Review Board Statement:** Not applicable.

**Informed Consent Statement:** Not applicable.

**Data Availability Statement:** Not applicable.

**Conflicts of Interest:** The authors declare no conflict of interest.

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
