# Peer review of "Toward Sustainable Development: The Causes and Consequences of Organizational Innovation"

_sustainability, doi:10.3390/su15108017_

Round 1

Reviewer 1 Report

1. In the abstract section, what do you mean by ESG'. Please extend the abbreviation. 

2. The first statement in the abstract needs evidence (reference), as well as line 71 in the introduction section.

3. In the introduction section, who is Schumpeter?

4. Please split the the introduction and add a literature review section. This kind of studies needs extensive literature review.

5. The literature review is not comprehensive and the range of reference is inadequate. 

6. The majority of the references are very old. Only a few references reflect contemporary research in the field of innovation and sustainable development. Please update the reference list and add more recent studies.

7. The main problem with the paper is its weakness of the literature review section, which does not display a complete review of all the topics under investigation. Please improve the literature review.  

8. The first statement in the conclusion section and the recommendations sections is not clear. Please clarify. What do you mean by explanatory power?

9. There is a great need to add a qualitative section to the methodology and analysis. The numbers represented are not enough in clarifying the whole view. Face-to-face feedback from the respondents will strengthen the argument for sure.

10. More focus on the meaning of sustainable development in the literature section is required.

Good luck in your revision. 

English language is more or less acceptable, the problem is that the tone of the paper is not academic as it should be. I recommend that a native academic to revise the paper before publication.

Reviewer 2 Report

This is an interesting article that contributes to the research on organizational innovation by means of the adoption of the ESG (environmental protection, social responsibility and corporate governance) approach. However, there are some issues that need attention. They are listed as follows.

1.       Please define ESG from the beginning. No every reader will be familiar with this acronym.

2.       The authors should explain in more detail the link between their research and the issue of sustainability which is the focus of the journal.

3.       In the research design section, add a brief introduction explaining the objective of this section and how it is organised.

4.       Please provide a brief and general explanation of the theoretical framework in Figure 1. I know that the elements of this framework are explained after this figure. But a general description with an explanation of the rationale behind this model is needed.

5.       In Section 2.3.1, the authors explain that creative personality traits in individuals was measured by means of the Creative Personality Scale. This scale was normalised to 1 for attributes that were positively associated with creativity and to -1 for attributes negatively associated with creativity. I think that this normalisation should be considered with caution because the individual impact of each attribute is not necessarily the same. This should be acknowledged as a limitation of the research.

6.       The sample used in the research contains 138 observations. This is a small sample implying that the results should be considered with caution. This has to be acknowledged as another limitation of the research.

7.       Please explain in the methodological section that Pearson correlation analysis and multiple regression analysis were used to determine the relationship between variables. Provide a justification for these techniques. Also explain that the stepwise regression and the hierarchical multiple regression techniques were employed for the analysis of the effects of individual, organizational, and environmental variables on organizational innovation. Provide a justification for these techniques.

8.       The authors state in the conclusions the following: “The findings of this study are in support of environmental and organizational variables having the largest explanatory power for organizational innovation, while individual creativity was found to have less of a correlation with organizational innovation, which is completely different to the findings of Amabile [18]”. I think this is a strong conclusion because the authors are basing their results in a small sample. The large exploratory power of environmental and organizational variables may also reflect a sample bias, and this should be acknowledged.

Reviewer 3 Report

Review report 

 Toward Sustainable Development: The Causes and Consequences of Organizational Innovation

The primary purpose of this study was to investigate the incorporation of relevant aspects of ESG into organizational innovation and further investigate it's influencing factors on innovation. The information electronics industry based at Hsinchu Science Park was selected to gather data for this study. However, the paper has methodological and technical flaws that must be corrected, and the following revisions are required to meet research rigor and enhance the quality of the article.

Introduction 

In the introduction, the authors should provide an overview of what is being discussed throughout the research. Please include the research significance, scope, and key objectives of this study. Only two references form relatively recent literature in the introduction, the reset most of the studies cited are outdated. Please update it is 2023 now. 

Literature review 

Apart from hypothesizing relationships, authors must explain each construct or variable in detail in the first section of the literature. This section has not been addressed. Please revise accordingly. Also, in the research design section, please provide theoretical justification for your proposed framework and discuss relevant underpinning theories.  

Methodology 

The formwork of the study shows direct and indirect relationships, but I don’t see any details on the SEM technique ? and the software used for analysis. It looks like you have used a regression technique. Please justify. The section on sample and population is also missing. How the sample was drawn from a population, and which sampling technique was used? Please elaborate and justify. 

Analysis and Results 

The output looks fine  but please summarize it. Please revise your analysis part. 

Discussion and findings 

In the discussion part we compare our study result with other studies. Please provide this section after the results. Please revise accordingly. 

Conclusion 

Looks fine 

References 

Please update accordingly.

Good luck 

Moderate changes required.

Reviewer 4 Report

This is a popular topic. Unfortunately, the authors provide little meaningful introduction to the results of existing studies. Instead of listing all the research hypotheses at once, the authors should conduct a detailed review of the existing literature before proposing each one.

Round 2

Reviewer 1 Report

The paper has surely improved after the first round of corrections. The authors provided an improved version of the theoretical background. They have also added on the references and provided a number of recent studies. The method and research design was also improved. In general, the overall structure of the paper was improved. I still do have to main concerns though; first, the paper needs strengthening in terms of English language. Many statements, sound informal and less academic. A native English reviewer needs to revise this final version of the paper. This is extremely recommended before publication. For instance, the first statement in the introduction: "With a growing global population, accelerating global development and increasing resource use and environmental 22 impacts, it is clear that the future will not be as sustainable in terms of choice and use as it has been in the past" is not yet clear, as well as many others in various locations. Please conduct another round of language editing by a specialist. The tone of the paper has to be academically strong. Second, there are still some statements that require referencing/evidence. For instance, "Although corporations are increasingly adopting ESG methods and regulations, little is known about how to incorporate ESG into business models" needs theoretical and empirical evidence. This statements represents the core of your research. Finally, please explain what do you mean by ESG in the Abstract section. You still did not fix it.

English language needs another round of amendment by a native reviewer.

Reviewer 3 Report

Point 5: Discussion and findings 

In the discussion part, we compare our study results with other studies. Please provide this section after the results. Please revise accordingly. This point is still yet to be addressed. 

Do proofread your work to enhance your English writing  quality 
